# Reporting of Freshwater Cyanobacterial Poisoning in Terrestrial Wildlife: A Systematic Map

**DOI:** 10.3390/ani12182423

**Published:** 2022-09-14

**Authors:** Alexandra K. Ash, Stuart Patterson

**Affiliations:** Royal Veterinary College, University of London, Hawkshead Lane, Hatfield AL9 7TA, UK

**Keywords:** cyanobacteria, cyanotoxin, wildlife, health, surveillance, poisoning, terrestrial

## Abstract

**Simple Summary:**

Harmful cyanobacterial blooms (cyanoHABs) have been reported globally, threatening human and animal health. They are encouraged by the warming climate and agricultural pollution creating nutrient-rich, warm environments, ideal for cyanobacterial proliferation. The cyanotoxins produced by these blooms have caused poisonings in many wildlife species; however, these cases are severely underreported, and many are likely missed. The aim of this systematic map was to collate, organise, and characterise all existing reports of cyanotoxin poisonings in terrestrial wildlife. We conducted a search of the published literature using online databases, yielding a total of 45 cases detailing incidents involving terrestrial wildlife. There is no current standard method for the reporting and diagnosis of cyanotoxin intoxication cases, and we provide recommendations on this to include both clinical diagnostic tools and investigative chemistry techniques. Less than half of all cases employed robust methods of detection and diagnosis based on our recommendations. Most cases were investigated after poisonings had already occurred, and only nine reports mentioned any effort to mitigate the effects of harmful cyanobacteria on terrestrial wildlife. This systematic map details terrestrial wildlife cyanotoxin intoxications from a diagnostic perspective, identifying how reporting can be improved, leading to more successful mitigation and investigative efforts in the future.

**Abstract:**

Global warming and over-enrichment of freshwater systems have led to an increase in harmful cyanobacterial blooms (cyanoHABs), affecting human and animal health. The aim of this systematic map was to detail the current literature surrounding cyanotoxin poisonings in terrestrial wildlife and identify possible improvements to reports of morbidity and mortality from cyanotoxins. A systematic search was conducted using the electronic databases Scopus and Web of Science, yielding 5059 published studies identifying 45 separate case reports of wildlife poisonings from North America, Africa, Europe, and Asia. Currently, no gold standard for the diagnosis of cyanotoxin intoxication exists for wildlife, and we present suggested guidelines here. These involved immunoassays and analytical chemistry techniques to identify the toxin involved, PCR to identify the cyanobacterial species involved, and evidence of ingestion or exposure to cyanotoxins in the animals affected. Of the 45 cases, our recommended methods concurred with 48.9% of cases. Most often, cases were investigated after a mortality event had already occurred, and where mitigation was implemented, only three cases were successful in their efforts. Notably, only one case of invasive cyanobacteria was recorded in this review despite invasive species being known to occur throughout the globe; this could explain the underreporting of invasive cyanobacteria. This systematic map highlights the perceived absence of robust detection, surveillance, and diagnosis of cyanotoxin poisoning in wildlife. It may be true that wildlife is less susceptible to these poisoning events; however, the true rates of poisoning are likely much more than is reported in the literature.

## 1. Introduction

Harmful cyanobacterial blooms (cyanoHABs) are becoming an increasing threat to human, animal, and ecosystem health [1,2,3]. CyanoHABs are caused by toxin-producing photosynthetic prokaryotes called cyanobacteria and are characterised by scums that can be green, bright blue, or other colours on the surface of brackish and freshwater. In cyanoHABs, typically one or two cyanobacterial species dominate the community [4,5], with major harmful genera being *Microcystis*, *Nodularia*, *Dolichospermum*, *Oscillatoria*, *Aphanizomenon*, and *Phormidium* [6]. In marine environments, these blooms are referred to as harmful algal blooms (HABs), dominated by toxic algal species and non-cyanobacterial species, such as those of the dinoflagellates and diatoms, resulting in fish, bird, and marine mammal mortality in extreme cases [7]. The eutrophication of freshwater systems encourages the formation of cyanoHABs by creating a nutrient-rich environment. Toxins produced by cyanoHABs are a diverse group of secondary metabolites, many of which can have radical impacts within the ecosystem [8]. The functional role of cyanotoxins is largely unknown; however, some studies suggest they may act as a chemical defence or physiological aide for cyanobacteria [8,9,10]. Cyanobacterial blooms can be harmful in a number of ways, such as by outcompeting phytoplankton, depleting oxygen, and producing cyanotoxins within the environment, killing fish and inhibiting plant growth in the process [4]. The most common toxins found in cyanoHABs include microcystins [11] and nodularins [12], which are hepatotoxins inhibiting protein phosphatase, and neurotoxins such as anatoxin-a [13], guanitoxin [14], 2,4-diaminobutyric acid (DAB) [15], and acute effects of ß-methylamino-L-alanine (BMAA) [16], which inhibit neuronal function. Although BMAA is labelled as a neurotoxin here, the chronic effects have been long debated [17,18,19,20,21]. These toxins threaten human health through the consumption of contaminated fish, molluscs, and gastropods, as well as coming into contact with bloom material during recreational water activities [4]. Whilst the threat of cyanotoxins to human and domestic animal health is not fully understood, resources and interest are invested in understanding public and domestic animal health risks. For wild animal health, the impact of cyanotoxins remains severely understudied, and wildlife health professionals currently lack the resources to ameliorate this knowledge gap. Mass mortalities from toxin poisonings have occurred within several protected areas, such as Kruger National Park in South Africa and Doñana National Park in Spain [22,23]. Without knowledge of the cyanobacterial species leading to these events and how to appropriately report them, investigations and mitigation will continue to be executed with limited knowledge or effectiveness. Understanding the mechanisms that induce cyanoHAB poisoning events in wildlife will aid in the design of effective surveillance policies.

Lots of uncertainty surrounds our current understanding of how cyanotoxins impact wild animal health and the mechanisms underlying such intoxications. It is unclear in some poisoning events if the ingestion of toxic algae alone leads to intoxication or if co-occurrence with plastic and other types of pollution exacerbates the toxicity. For example, cyanotoxins have been observed to act in concert with other toxic compounds [24,25,26]. In some cases, cyanotoxins have been listed as a contributing factor alongside heavy metals and pesticides in wildlife mortality [27]. It has also been suggested that cyanotoxins can cause diseases such as avian vacuolar myelinopathy in American coots and bald eagles [28,29]. Consequently, cyanotoxins may be the etiological agent of disease, cause intoxication, or act in combination with other pollutants. Further, cyanotoxins are known to bioaccumulate in fish and gastropods and cause biomagnification in humans, with a range of evidence supporting the accumulation of cyanotoxins in aquatic species [30,31,32]. However, few studies have detailed these chronic toxic effects in terrestrial wildlife, such as Wöller-Skar et al. [33] and Cox, Banack, and Murch [34], who described the bioaccumulation of BMAA and microcystin within bats. Without further research into the direct causal mechanisms of morbidity and mortality related to cyanoHABs and cyanotoxin poisoning, appropriate mitigation and diagnosis cannot be developed for wildlife.

Wildlife health surveillance is often absent in cyanotoxin investigations, and there is currently no standard for the diagnosis or detection of cyanotoxin poisoning in wildlife. Without knowledge of the pollutants present during a poisoning event, the cyanobacterial species involved, the toxin involved, the surrounding landscape use, or weather events, we cannot identify those wildlife species that are at the most risk of intoxication. Often, the presence of toxic cyanobacteria is detected only when a cyanoHAB is in full force, after animals have already been exposed to the toxins [4,7,35]. Toxic cyanobacteria are already present within a lake system prior to bloom formation [36] and establishing their presence using investigative chemistry techniques could help to anticipate and monitor poisoning events. With no standard method of reporting these cases in place, we are unable to identify possible mechanisms underlying these poisoning events. As a result, the characterisation of species susceptibility and the design of effective species-level or ecosystem-level conservation strategies in response to these events are unable to be carried out. However, these major gaps in knowledge are largely due to insufficient surveillance and poor detectability of mortality in wildlife, which is often challenging to overcome [37]. There is also a lack of evidence for poisonings in cryptic species such as amphibians and reptilians, which are known to be difficult to detect [38]. Interestingly, sources have cited cyanotoxins as a potential source of population decline for amphibians [39], but there are no dedicated surveillance schemes to elucidate this. It is recognised that cyanotoxins can have an impact on terrestrial wildlife health, with reports likely to become more frequent with increasingly hot and dry climates [40]. Due to the lack of survey efforts in the surveillance of cyanotoxin poisonings, it is evident that more research is needed in order to support and facilitate the detection and diagnosis of wildlife cyanotoxin poisonings.

This systematic map aims to collate, describe, and catalogue the available evidence relating to terrestrial wildlife mortality and morbidity associated with cyanoHABs. In particular, we focused on studies that reported terrestrial wildlife casualties mostly occurring in and around freshwater. No systematic reviews, to our knowledge, have detailed cyanotoxin intoxications in the sole context of terrestrial wildlife due to freshwater cyanobacteria. This synthesis is principle in understanding how cyanotoxin poisonings in terrestrial wildlife are investigated and reported, if environmental contaminants exacerbate the frequency of poisoning events, what mitigation methods are effective, and how we might improve these investigations.

## 2. Protocol 

### 2.1. Literature Sources and Search Terms 

The following systematic map adhered to guidelines outlined by the CEE and PRISMA [41,42]. To identify relevant literature, a systematic search of the databases PubMed and ISI Web of Science was conducted using two sets of relevant search terms corresponding to acute and chronic intoxications. Two separate searches were conducted, as bioaccumulation and acute toxicity are reported and investigated differently, often published and conducted within different fields of research. Grey literature was included within the review through the captured results, but due to resource constraints, a targeted search for grey literature was not conducted. Below are two sets of search terms created for cases of acute intoxication and bioaccumulation of cyanotoxins, respectively:

(ALL = (cyanobacteria*) OR ALL= (harmful algal bloom*) OR ALL= (blue-green alga*) OR ALL = (bloom toxicity)) AND (ALL = (wildlife) OR ALL = (mammal*) OR ALL = (bird*) OR ALL = (amphibian*) OR ALL = (invert*) OR ALL = (reptile*));

(ALL = (cyanobacteria*) OR ALL = (harmful algal bloom*) OR ALL = (blue-green alga*)) AND (ALL =(bioaccumulat*) OR ALL = (biomagnificat*) OR ALL=(biontransfer*)).

Both sets of search terms were searched within PubMed and ISI Web of Science, and results were combined for the screening stage. Despite the primary focus of this review being cyanoHABs, the inclusion of the term “harmful algal bloom” was necessary to capture cases where terrestrial wildlife was intoxicated by marine algal species and non-cyanobacterial species.

### 2.2. Literature Search and Inclusion Criteria 

The results of searches were managed in EndNote (version X9, Clarivate), which was also used for removing and identifying duplicate studies. The primary author conducted the literature searches, screening, and extraction of data. Two search methods were used for inclusion: the use of scientific literature databases and subsequent snowballing. After database searches were conducted, the collected papers were screened for inclusion based on the following listed criteria:Literature reporting cyanotoxin poisonings in terrestrial wildlife, including reptiles, amphibians, mammals, birds, and aerial insects with aquatic larval stages. Cyanobacterial poisonings in other populations, such as aquatic species, domestic species, and humans, were excluded.Literature discussing terrestrial wildlife exposure to cyanobacterial species and toxins, including direct or indirect exposure and including chronic or acute intoxication. Literature discussing poisonings due to experimental exposure to toxins was excluded from this review.Literature discussing algal poisoning in the context of coastal, marine, or estuarine waters (HABs) was only included if terrestrial wildlife species were affected. In these instances, non-cyanobacterial algal species and algal toxins may be implicated. Due to the focus of the review, it is relevant to include these cases in investigating surveillance techniques for terrestrial wildlife poisoning.Grey literature was not actively searched for evidence of poisonings but was not excluded if captured by the literature search.We included only literature in English, but limitations regarding this are acknowledged, as cyanoHABs may occur anywhere in the world and be reported in any language. Due to resource and time constraints, we were not able to include any non-English language studies. Any texts excluded due to language were recorded.All study types except for reviews were included.

An adapted version of Wohlin’s [43] snowballing method was performed after database results were screened against the inclusion criteria. Snowballing involved the screening of references in each included paper, following the same method for inclusion as above. This occurred alongside the initial screening process and ensured no poisonings were omitted, providing a comprehensive representation of the existing literature.

### 2.3. Recommended Method of Diagnosis and Investigation 

In the absence of a standardised method, we developed our own recommendations based on the current literature detailing methods of diagnosis and reporting for human cyanobacterial poisonings. Based on the existing literature, our methods included the sampling and detection of cyanotoxins via investigative chemistry; the identification of cyanobacterial species involved; evidence of exposure to toxins through observation or gut content analysis; and evidence of toxicosis or bioaccumulation through post-mortem examination (PME), histopathology, and/or clinical signs [44,45,46]. For the targeted screening of toxins, liquid chromatography–tandem mass-spectrometry (LC-MS/MS) is recommended, especially where both native and isotopically labelled standards are usable [44,47,48,49,50]. Further, liquid chromatography with high-resolution mass spectrometry (HRMS) or time of flight (TOF) is best at detecting unknown toxins or those for which standards are not readily available [44,47,49,50]. An alternative to these methods would be enzyme-linked immunosorbent assay (ELISA), where kits are commercially available for some toxins (e.g., microcystins, anatoxin-a, cylindrospermopsin, and saxitoxin) [44,47,49,50]. Considering the application and feasibility of these techniques for wildlife health surveillance and the remoteness of some cases [32,36,38,39], it is likely that methods such as ELISA are more practicable where accessible labs do not have the facilities necessary for liquid chromatography and mass spectrometry techniques. Samples for toxin screening may be obtained from water bodies, soil, or affected tissues from target organs [47]. Water samples are the most reliable medium to assess toxin presence and concentration, as water matrices were used in the development and standardisation of detection techniques; however, for the purpose of diagnosis of intoxication in wildlife, obtaining tissue samples is recommended if feasible [47]. Additionally, the identification of the cyanobacterial species involved should always be attempted. Previous methods recommend light microscopy to identify cyanobacterial species, but it is well known that toxic and non-toxic counterparts of the same species look identical and are unable to be distinguished through microscopy [46]. Therefore, genetic methods such as PCR are recommended for the identification of species, as they are able to identify multiple species simultaneously and accurately. PCR enables rapid, cheap, and reliable identification and surveillance of toxic cyanobacterial species present in water bodies. Our recommended methods were used as a coding variable, in which we compared methods of reporting and diagnosis of poisonings in studies to our own recommendations.

### 2.4. Data Extraction

Data were extracted from each study to answer the main aim. The meta-data of each article, such as year, author, study type, country, outcome, and toxin or cyanobacterial species involved in each incident, were extracted and are listed in Table 1. Additionally, author affiliation was coded to determine collaboration between different fields of research and catalogue the variety of skillsets employed within each investigation. This can be used as a tool to assess where more collaboration may be needed, as wildlife health surveillance combined with cyanoHAB surveillance requires a multi-disciplinary approach.

A set of coding variables were additionally developed for the investigative methods employed in each case. These were generally characterised as retrospective or active, where retrospective investigations represented the absence of wildlife health surveillance, and active investigations indicated the presence of surveillance. Surveillance was defined as any effort to monitor wildlife health, whether established or casual, and included citizen science survey efforts, cyanoHAB surveillance, other disease monitoring programmes, and water or sample collection by academics. Exposure was classified as acute or chronic, as detailed in a review conducted by Young et al. [51]. Acute exposure was described as a single exposure event, and chronic exposure was described where exposure was measured over time. If studies did not detail chronic or acute exposure, then based on the time of exposure, a decision was made by the author. If outcomes and symptoms occurred 28 days after exposure, the case was labelled as “acute,” and after 28 days, it was labelled as “chronic” [52], as symptoms are more likely to dissipate after 28 days in acute cases [53]. Each paper was compared against our recommended method for investigation based on recommendations in the prior subsection. The coding variables used to extract and record this information are listed below in Table 1.

### 2.5. Data Synthesis and Visualisation 

Data were visualised qualitatively and summarised narratively. Microsoft Excel (ver. 16.16.27) was used to produce descriptive statistics using extracted meta-data illustrating the year, study type, country, outcome, animal species, cyanotoxins, and cyanobacterial species involved. R version 4.1.1. [54] was used to produce visualisations using the packages ggplot2 [55] and dplyr [56]. EviAtlas [57] was additionally used to visualise the systematic map database and produce a map with extracted meta-data.

## 3. Results

### 3.1. Eligibility of Studies 

The movement of information through different phases of the review process is illustrated by the PRISMA diagram in Figure 1. Reports excluded for study type included reviews detailing animal exposure to cyanotoxins. Reports excluded due to the population detailed cases in humans, domestic animal species, or non-terrestrial wildlife species. Reports excluded due to exposure primarily included experimental exposure to cyanotoxins. Lastly, reports excluded due to context were largely those detailing exposure to other environmental toxins independent of cyanotoxin poisonings, such as heavy metals and pesticides.

### 3.2. Characteristics of Included Studies 

There were 45 publications included in this review (Table A1). Table 2 provides a narrative summary of important characteristics of all included papers. Publications increased over time, as 40 (Figure 2) papers were published from 1990 onwards, compared to only 5 prior to 1990. The year in which cases occurred (Figure 3) does not follow the same trend as publishing, as there is a time lag from when cases occurred to when they were reported. The presence of a harmful cyanobacterial bloom was not a requirement for inclusion, but they were present in 29 (64.4%) cases. Benthic cyanobacterial mats also contributed to intoxication, implicated in four (8.9%) cases, with two (4.4%) cases describing the poisoning of terrestrial wildlife by non-cyanobacterial species. Only one case reported the cause of intoxication to be an invasive species of cyanobacteria, a species of the Stigonematales order. The locations of all cases were mapped using EviAtlas (Figure 4).

### 3.3. Lead Author Affiliation 

University departments constituted the majority of primary author affiliations, writing 60.0% of reports (Figure 5). The number of institutions in each study ranged from one to seven, with 23 (51.1%) reports collaborating with institutions in different fields.

### 3.4. Cyanobacterial Species and Toxins

The most abundant cyanotoxins and cyanobacterial species were microcystins and *Microcystis* spp. Summaries of species and cyanotoxins implicated in cases are provided in Table 2. The co-occurrence of toxins occurred in 20 (44.4%) cases, with an average of 1.89 toxins occurring in each case. Investigations not using microscopy or PCR could not identify the cyanobacterial species involved, so species was unknown in 13 (28.9%) cases. Additionally, only one case, in South Carolina, United States, reported an invasive epiphytic cyanobacterial species of the order Stigonematales.

### 3.5. Animal Species Involved and Exposure 

Avian species constituted the majority of cases (33.3%), and a summary of the species involved is provided in Table 2. Five cases involved both wild and domestic animals. The species of domestic animals affected alongside wildlife included domestic dog *Canis lupus familiaris*, chicken *Gallus gallus domesticus*, cow *Bos taurus*, horse *Equus caballus*, pig *Sus scrofa domesticus*, turkey *Meleagris gallopavo* f. domestica, goose *Anser* sp., cat *Felis catus*, and sheep *Ovis aries*. The majority of studies reported acute exposure (64.4%), while chronic exposure (31.1%) was reported less often, and exposure was unknown in some cases (6.7%). Of the 14 chronic cases reported, 100% of these cases reported the bioaccumulation or biotransfer of cyanotoxins in wildlife.

### 3.6. Surveillance and Diagnostics 

Cases were classified as retrospective or active investigations, indicating the approach of the investigation. Retrospective investigations constituted 31 (68.9%) cases, and active ones were implemented in 14 (31.1%) cases. All cases involving chronic exposure were investigated actively. Of these active investigations, 11 cases were discovered through academic researchers investigating the accumulation of toxins, 1 was discovered through routine water monitoring, 1 was found when monitoring reintroduced white rhinoceroses *Ceratotherium simum*, and 1 was encountered while investigating the drivers of acute mortality in wildlife. Additionally, where a bloom did occur (*n* = 29), 65.5% of these cases were investigated retrospectively. In cases where mass mortality occurred (more than 10 individuals, *n* = 25), 92% of these cases were also investigated retrospectively.

A diagnosis was made in 25 cases, and our recommended methods of diagnosis agreed with the analytical methods used in 76.0% of these cases (48.9% of cases overall). This recommendation was used to assess the quality of reporting involved in each case. A combination of clinical diagnostic methods was used in 42.2% of cases, as recommended by our methods. Post-mortem examination was employed most often, followed by tissue samples for toxin detection, histopathology, clinical signs, and blood samples (Table 3). The number of clinical diagnostic methods used in each case varied from 0 methods (22.2%) to 4 methods (2.2%), with an average of 1.38 diagnostic methods used. Microscopy or genetic methods are necessary for the identification of cyanobacterial species, and only 19 (42.2%) and 3 (6.7%) cases, respectively, used these methods (Table 3). Bioassays were employed in 12 (26.7%) cases, and the majority were mouse bioassays. One case [58] used a cow bioassay to determine the toxicity of bloom material, where bloom material was administered orally to a cow, and clinical signs were observed. A summary of all other investigative chemistry techniques used in cases is provided in Table 3.

## 4. Discussion

Highlighting limited resources in ensuring adequate detection, diagnostic, and surveillance techniques, the 45 cases included in this review represent our current knowledge of terrestrial wildlife cyanotoxin poisonings. Of all 45 cases, 48.9% mirrored our recommended methods of diagnosis using a combination of investigative chemistry techniques and clinical diagnostic tools. Mitigation was implemented in only nine cases, of which only three were successful in their efforts. Additionally, active investigations constituted 31.1% of all cases, meaning most poisonings were discovered after mortality has already occurred. Notably, only 44.4% of cases used microscopy and genetic methods to identify cyanobacterial species. Despite the knowledge that invasive toxic cyanobacteria exist throughout the globe [59,60,61], only one species in this review was identified as invasive. This could prove highly problematic if the misclassification of invasive species is occurring within investigations.

### 4.1. Invasive Cyanobacteria 

Interest in invasive cyanobacteria has gained momentum in the last decade of research, e.g., [62,63,64,65,66,67,68], and the lack of reports for these species implicated in wildlife casualties could be concerning. Only one invasive species was observed in this review, an epiphytic species of cyanobacteria in North America [69]. Although 28.9% of cases did not attempt any genetic or microscopy methods to identify cyanobacterial species, the majority of these cases occurred prior to 2010. Thus, the absence of relevant knowledge on invasive species at the time of investigation likely plays the largest role in preventing the correct identification and categorisation of invasive species. It is known that the global movement and trade of goods has facilitated the spread of toxic *Microcystis* and *Dolichospermum* species via shipping ballasts in the Great Lakes of America [70]. Whilst the transfer of freshwater cyanobacterial species may not be a regular occurrence via shipping ballasts, increased traffic across larger water bodies may encourage the spread of invasive toxic cyanobacteria, as has been documented in the Great Lakes. Additionally, previous studies have already highlighted the importance of understanding population dynamics and species composition to predict the occurrence of invasive species and cyanoHABs [1,71]. With the addition of new detection methods and surveillance strategies, new cyanobacterial species and toxins are being discovered rapidly [72,73]. To this end, as investigators continue to report wild animal cyanotoxin intoxications, it is important that they conduct a thorough search of available analytical methods and current taxonomic names for cyanobacteria. However, with a lack of baseline data, invasive cyanobacteria have the potential to proliferate unknowingly even if the correct analytical methods are employed.

### 4.2. Co-Occurrence of Cyanotoxins and Environmental Contaminants 

Co-occurrence of cyanobacterial species occurred in 44.4% of cases, but due to a lack of identification through microscopy or PCR, this is likely an underrepresentation. As some cyanotoxins act synergistically, co-occurrence can be detrimental [24,74]; this underlines the advantages of simultaneously detecting multiple species using genetic methods [75]. Investigations tend to emphasise toxin identification due to resource constraints, as this informs treatment options. ELISA was used most often, followed closely by HPLC, to identify toxins, and this may indicate resource limitations for cyanotoxin investigations involving wildlife casualties. Methods such as HPLC and mass spectrometry require skilled lab technicians and expensive equipment [44], whereas commercially available ELISA kits are rapid and inexpensive [44]. As it stands, no case stated resource constraints as a limitation, however the absence of robust surveillance and diagnosis methods stresses the possibility of resource constraints within wildlife health surveillance, preventing detailed and rigorous investigations from being carried out.

Further, the co-occurrence of cyanotoxins could exacerbate the bioaccumulation of toxins. All cases of chronic exposure were characterised as bioaccumulation, occurring within 31.1% of cases. This might indicate miscategorisation of bioaccumulation cases, as animals may often be expressing sub-lethal effects instead [76,77]. Heavy metals have the potential to propagate throughout food webs, inducing wildlife poisoning [78], and cyanotoxins harbour the same potential. In this review, the transfer of cyanotoxins from one food web to another occurred from insects to bats, bats to humans, frogs to birds, and more [34,79]. With a wide variety of species implicated in the accumulation of cyanotoxins, this issue is not restricted to one habitat or even one ecosystem. It has been argued within the literature that microcystins cannot biomagnify within food webs, but rather transfer or accumulate throughout the food web [80,81]. This is an important distinction, as biomagnification, bioaccumulation, and biotransfer are often used interchangeably but describe different processes.

Other toxic compounds may act alongside cyanotoxins to cause wildlife casualties, including heavy metals, pesticides, and organophosphates [82,83,84,85,86]. All eight cases involving the co-occurrence of environmental contaminants were acute intoxications leading to death. Only one case determined the implication of sewage pollution in cyanoHAB formation [87], thus contributing to the mortality of wild birds in this one case. However, no cases involving environmental contaminants stated the direct contribution of pollutants to the mortality or morbidity of animals, only the facilitation of bloom formation. The possibility of cyanotoxins co-occurring with environmental contaminants and both contributing to casualties is still debated within the existing literature [24,26,80,86,87]. The majority of reports do not note the presence of environmental contaminants nor attempt to identify the presence of any pollutants. It is possible that pre-existing conditions related to chronic exposure to other non-cyanobacterial contaminants facilitated the susceptibility of these animals to disease, which has been recorded in the North Sea with heavy metal pollution in marine mammals [88], in frogs with pesticide exposure [89], and in avian species with air pollution [90]. Further, water bodies polluted with pesticides and heavy metals are often subjected to agricultural runoff, a known driver of bloom proliferation [24]. Yet, with many cases failing to recognise the potential impacts of other contaminants such as immunosuppression and physiological stress on wildlife poisonings, the complex interactions between biotic and abiotic factors in these cases remain unknown. Further investigations into the co-occurrence of these compounds are needed to substantiate arguments made in the literature.

### 4.3. Methods of Surveillance and Diagnosis

Adequate methods for the detection of cyanotoxins and the monitoring of wildlife diseases are the largest barriers to overcome when investigating cyanotoxin poisoning in wildlife [91,92,93]. In cases where mass mortality occurred, most (92.0%) were investigated in the absence of either formal or casual wildlife health surveillance. Dedicated wildlife surveillance schemes for cyanotoxin intoxication do not exist as they do for public health [94,95]. Additionally, threshold values for short-term exposure to microcystin-LR were published by the World Health Organization (WHO) in 2021 for drinking water [96]; however, these values do not exist for wildlife, even though microcystins contributed to the majority (84.7%) of all wildlife cases in this review [54]. Wildlife poisonings can act as a sentinel for public health [53], and a joint surveillance effort would benefit both humans and animals. Surveillance occurs on a regular basis for public health in the Great Lakes of North America and Lake Taihu of China [97,98] and may work well with recommendations set out by Brookes et al.’s [95] guide to monitoring toxic cyanobacteria. From our results, wildlife cyanotoxin poisoning cases are typically discovered in the absence of surveillance, and monitoring programmes could be integrated conveniently with public health objectives to combat this.

Further, toxic and non-toxic counterparts of cyanobacteria are not easily distinguishable via microscopy [99]. PCR and qPCR are proposed methods to identify toxic cyanobacterial species simultaneously [44,99,100,101,102], but only 3 cases used PCR compared to 19 using microscopy. Although neither the U.S. Environmental Protection Agency (EPA) nor any other government agency has identified a gold standard for the detection of cyanotoxins [44], they do provide recommended toxin detection methods, such as those recommended in this manuscript [103]. They also acknowledge the scope of each toxin detection method and provide information on the best methods for known cyanotoxins [103]. Additionally, the EPA comments on the widespread use of commercially available and inexpensive ELISA kits [103]. As recommended methods do exist, resource constraints within the field of wildlife health surveillance provide a better explanation for only 48.9% of investigations aligning with our suggested methods. Unofficially, investigative chemistry techniques are recommended as a gold standard because they are highly sensitive [44] despite being expensive. Kaushik and Balasubramanian [44] endorse a multi-method system of analysis, including initial species identification via microscopy and then toxin analysis via LC-MS. Recommended methods such as LC-MS/MS, LC with HRMS or TOF, or ELISA to identify toxins implicated and their concentrations are common and widely accepted, with LC-MS/MS labelled as the unofficial gold standard for toxin identification [44,47,49]. Whilst ELISA has been labelled as “semi-quantitative” [47], it is the most practicable and cheap method available where known toxins are implicated. For the purposes of wildlife health surveillance, we recommend the use of commercially available kits for microcystins, nodularins, anatoxin-a, cylindrospermopsin, and saxitoxin. However, it is important to note that recommendations in the literature were largely published within the last decade [44,45,46,47], and over half of all cases in this review were published prior to 2010. It is acknowledged that these investigations did not have access to current tools or recommendations to carry out robust investigations, and the best recommendation for current investigations is to remain informed of the most appropriate analytical methods to date. Consequently, methods such as those proposed here, by Kaushik and Balasubramanian [44], and by Pacheco, Guedes, and Azevedo [102] equip researchers with more robust analytical methods that allow for cheap and simultaneous detection of multiple cyanobacterial species and toxins.

Whilst there is an understanding of ideal methods for identifying cyanobacteria and cyanotoxins, currently, access to resources, training, and funding for such investigations remains a limitation in many wildlife health surveillance schemes [104]. Laboratories and resources available to conduct such analyses are difficult to access in developing countries, where impacts may be the most detrimental [105], and veterinary staff are not equipped with the training to identify cyanobacterial species and toxins. This may explain why developing countries have not formally reported many wildlife poisonings, despite previous research suggesting that blooms occur in these regions most often [105]. As improvements to investigations are central to the outcomes of this review, resource limitations must be addressed and overcome in order to progress methods of wildlife health surveillance (WHS). Considered from a One Health perspective, cyanotoxins threaten not only wild animal health but also domestic animal, ecosystem, and human health. Spotlighting wildlife intoxications from cyanoHABs in this way can provide a gateway to more interest, funding, and support.

### 4.4. Improving Investigations 

If resource limitations can be addressed, we can move on to how an ideal investigation should be carried out. Where acute intoxication occurs, appropriate surveillance methods and mitigation should be prioritised, as only a limited number of studies do so. Figure 6 depicts the flow of wildlife cyanotoxin investigations, suggesting the necessary steps of surveillance, discovery, diagnosis, and mitigation.

Targeted WHS is crucial for habitats and water bodies where cyanoHABs and poisonings are known to reoccur. As toxic cyanobacteria are always present within their environment, the levels of these species will need to be monitored before they can reach a harmful threshold, becoming a wildlife health risk. The aim of this hazard-specific health surveillance would be to detect wildlife casualties from cyanotoxins but also the presence of potentially toxic and invasive cyanobacterial species. With the intrinsic sentinel benefits of detecting wildlife poisonings, collaboration between human, domestic animal, and wildlife health professionals is essential to combat the difficulties in detection. Only half of all papers in this review allowed for the collaboration of professionals from different fields. These investigations require more integration and collaboration between climate scientists, marine and freshwater ecologists, public health professionals, veterinary health professionals, and remote sensing technology experts. With many cases involving large fauna, such as artiodactyls and carnivores, remote sensing techniques are a valuable tool in detecting poisonings. This would enable monitoring of the frequency and intensity of cyanoHABs, particularly in areas where mass mortalities have occurred, such as the lesser flamingos *Phoeniconaias minor* in Kenya [25,106]. Additionally, as cyanoHABs are linked to climatic fluctuations, coordinating reports of wildlife poisonings among countries would be beneficial in understanding the scale of the impact, as cyanoHABs are likely happening at a regional scale.

Satellite-derived information on the land cover type, temperature, humidity, spectral data, and other environmental variables can be used in the surveillance of algal blooms in fresh, brackish, and salt water [107,108,109,110,111,112,113,114]. Formal surveillance programs are already in place within the United States, such as the Center for Disease Control’s (CDC) One Health Harmful Algal Bloom System (OHHABS) and the Environment Protection Agency’s (EPA) Cyanobacteria Assessment Network (CyAN). These surveillance schemes work in partnership primarily with the health departments of states and territories but also with animal health officials, researchers, and members of the public. Backer et al. [108] reviewed CDC algal bloom surveillance data from 2007–2011 but provided little elaboration or detail of wildlife casualties. Whilst these schemes make an effort to monitor wildlife casualties alongside human and domestic animal illnesses, Backer et al. [108] emphasised the routine monitoring of public recreational waters, neglecting water bodies solely used by wildlife. As the routine monitoring of isolated water bodies for wild animal health is inherently difficult due to access, funding, and feasibility, this can be overcome with the use of unmanned aerial vehicles (UAV) [113]. With these surveillance schemes, there is always a time lag, and although challenging to succeed, it can also be ameliorated with the use of remote sensing technology. With the technology, data, and expertise already in place, cyanoHAB surveillance must prioritise wildlife health alongside human, domestic animal, and ecosystem health.

Next, the identification of the cyanobacteria and cyanotoxin implicated in the event is paramount in understanding the wider implications of the event, such as invasive cyanobacterial species. Genetic methods, such as PCR and qPCR, can be used for the identification of toxic cyanobacterial species present [114]. Careful consideration in determining what investigative chemistry technique to use (e.g., LC-MS/MS, LC with HRMS or TOF, and ELISA) and its limitations must be acknowledged prior to investigation. Toxin and cyanobacterial species can be identified through water samples but also through gut or faecal contents and fur or feathers where the animal has been in contact with bloom material [115]. A diagnosis of poisoning can be made using a multitude of methods, provided that evidence of ingestion or contact with bloom material is present. It is also important to understand the combined effects of environmental contaminants, infectious disease, and cyanotoxins. Kotut and Krienitz [116] discussed cyanotoxins suppressing the immune function of lesser flamingos, making the animals more vulnerable to infectious disease, and Papadimitriou et al. [117] discussed other abiotic and biotic factors alongside cyanotoxins contributing to the mortality of dalmatian pelicans *Pelecanus crispus*. This would require a more thorough investigation into all environmental contaminants and infectious diseases that wildlife is exposed to in addition to cyanotoxins. Evidently, the diagnosis of cyanotoxin poisoning is not so obvious when considering the additional impacts of other environmental and disease stressors.

The last step in these investigations should include the consideration of mitigation methods that aim to prevent future intoxications in wildlife and subsequently in domestic animals and humans. Many mitigation methods employed in wildlife cases are context-specific, such as dam drainage and burning of grazeland to protect megaherbivores within Kruger National Park [23,118,119]. However, these methods were only temporary, and poisonings persisted once mitigation was no longer maintained. The majority of cases in this review neglected to mention or apply any mitigation to prevent further poisonings, which remains problematic to the progression and success of these investigations. These methods can largely be divided into six groups: water management and treatment; translocation of animals; habitat management; disease surveillance; education and outreach; and clinical treatment. When deciding mitigation efforts to put in place, each one of these six mitigation themes must be considered, although which methods are chosen to be implemented will be largely context- and resource-dependent. Mitigating against cyanobacterial poisoning in wildlife is challenging; however, it requires consideration in every case to help progress our understanding and management of poisonings.

### 4.5. Review Limitations 

This study provides a thorough review of the current literature surrounding cyanotoxin poisonings in terrestrial wildlife; nonetheless, limitations do exist. A comprehensive search strategy was adopted, but in the absence of additional databases for grey literature, the breadth was limited. It is recognised that only including publications in English was the largest limitation of this review [120], but resources were not available to expand literature searches to other languages.

## 5. Conclusions

This systematic map outlines reports of cyanobacterial poisoning in terrestrial wildlife to date and which methods of detection, reporting, and diagnosis were employed in each case. Ultimately, whilst we have information on the surveillance, diagnosis, and mitigation of cyanotoxin poisonings in wildlife, which are key to robust investigations, these are not often fully employed. Addressing resource constraints in cyanoHAB surveillance should be a priority in wild animal health, especially in developing nations where cyanoHABs are underreported and may be occurring often. Most papers in this review were published after 1990, and methods of investigating wildlife poisonings are rapidly developing. However, with current projections of climate change, harmful cyanobacterial blooms will continue to proliferate, and poisonings will consequently persist. The progression of investigations in this field needs to mirror the rapidity of the changing climate and technological advances. The lack of identification of cyanobacteria to the species level is problematic and tackling this involves inexpensive methods such as PCR and qPCR, as well as investigators informed on updated taxonomic names. In conclusion, targeted research, increased funding and resources, and appropriate methodology can improve our understanding of cyanotoxin poisonings in wildlife to benefit human, animal, and ecosystem health synergistically.

## Figures and Tables

**Figure 1 animals-12-02423-f001:**
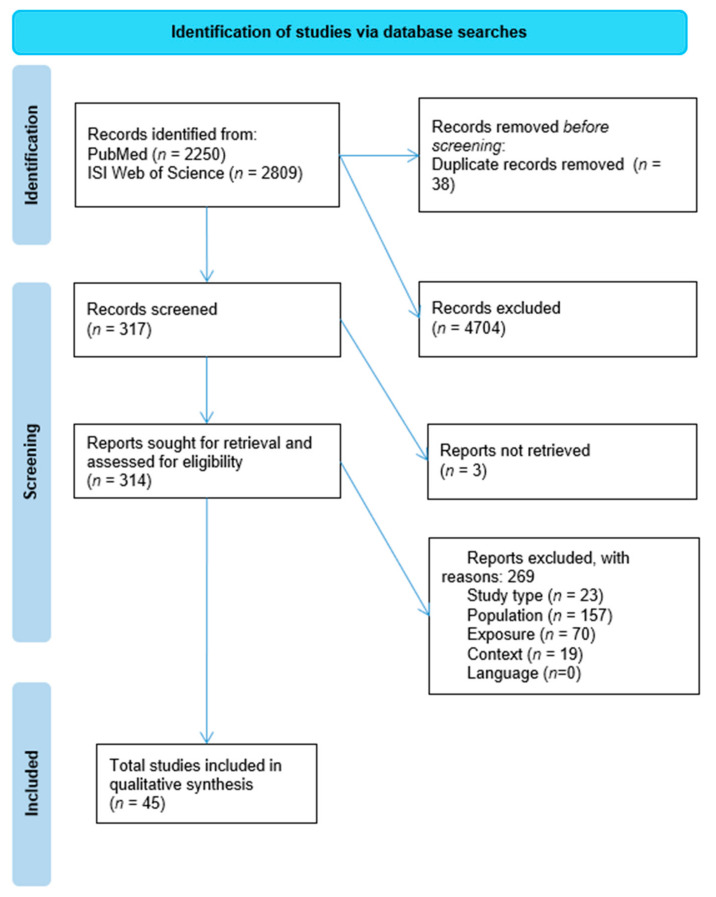
PRISMA diagram illustrating the flow of information through each screening step.

**Figure 2 animals-12-02423-f002:**
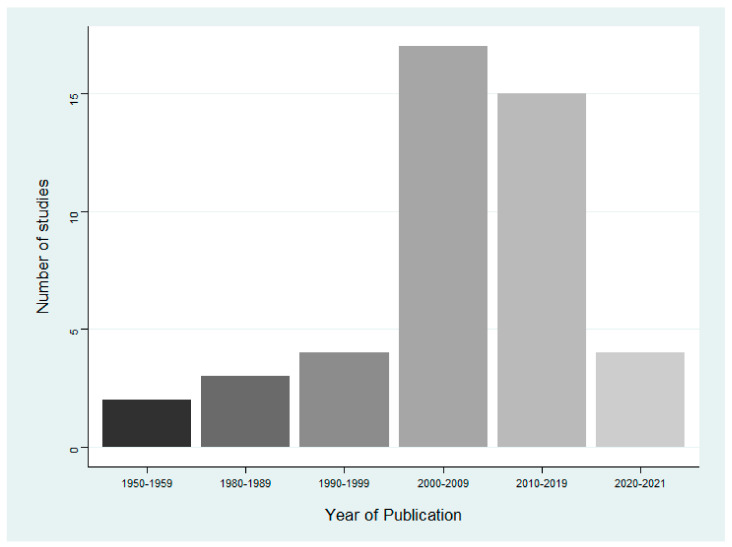
Number of publications of cyanotoxin reports in terrestrial animals in each decade, *n* = 45.

**Figure 3 animals-12-02423-f003:**
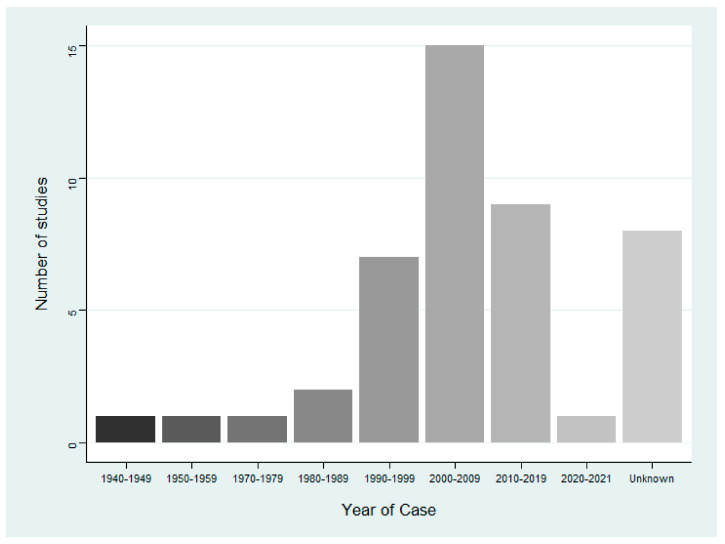
Number of cases of cyanotoxin intoxications in terrestrial animals in each decade, *n* = 45.

**Figure 4 animals-12-02423-f004:**
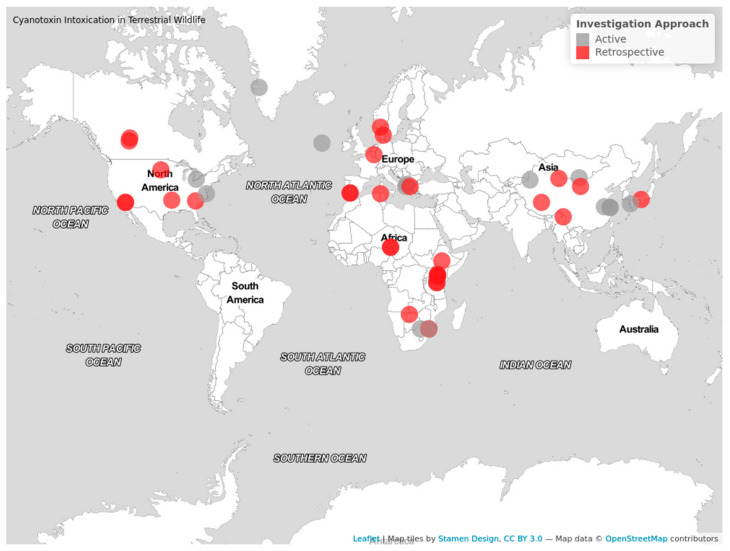
A map displaying the locations of all included cases; red circles represent cases investigated retrospectively, and grey indicates cases investigated actively. Produced using the EviAtlas Tool [57].

**Figure 5 animals-12-02423-f005:**
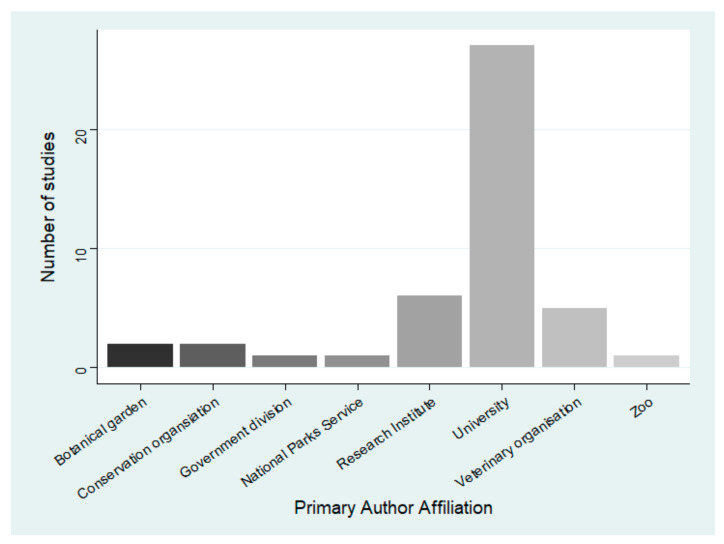
The lead author affiliation of reports on wildlife cyanobacterial poisoning.

**Figure 6 animals-12-02423-f006:**
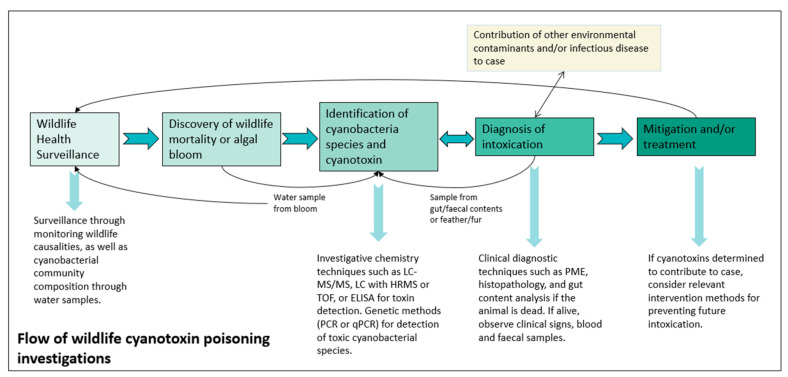
Flow chart of wildlife cyanotoxin poisoning investigations.

**Table 1 animals-12-02423-t001:** Meta-data coding variables and description.

Category	Coding Variable	Description
Meta-data	Reference	Author, date, title, and publisher
	Author affiliation	What organisation the authors are affiliated with (veterinary institute, university, government, conservation organisation, etc.)
	Date of publication	Year of publication
	Date of poisoning event/study	Year of event
	Study type	Experimental, observational, survey, review, etc.
	Location of study	Country and continent of study
	Outcome	The outcome resulted in death or recovery
	Case type	Chronic or acute intoxication
Methods of reporting and diagnosis	Cyanobacterial species	Species of cyanobacteria involved in event
	Toxin	Toxin involved in event
	Identification of toxin and cyanobacterial species	Investigative chemistry techniques used to identify cyanobacterial species and toxin (high-performance liquid chromatography, enzyme-linked immunosorbent assay, protein phosphatase inhibition assay, bioassay, mass spectrometry, etc.)
	Presence of a bloom	Was a cyanobacterial bloom present? (yes/no)
	Environmental conditions	The environmental conditions present (hot and dry, windy, prolonged sunshine, etc.)
	Bloom frequency	The frequency in which blooms are occurring
	Environmental contaminants	The environmental contaminants present at time of poisoning (sewage, fertiliser, heavy metals, pesticides, etc.)
	Diagnosis	Was an official diagnosis made in the case? (yes/no)
	Diagnostic methods used	Types of methods used to reach a diagnosis (post-mortem examination, histopathology, blood chemistry, clinical signs, stomach/faecal contents, etc.)
	Recommended method of investigation	Did the investigation follow our recommended method? (yes/no)
	Exposure pathway	How was the animal exposed to the toxin (ingestion, inhalation, transplacental transfer, transdermal uptake, etc.)?
	Number of carcasses	Number of carcasses discovered post-poisoning event
	Animal species	Animal species involved in the poisoning event
	Investigation approach	Was the investigation active or retrospective?
	Mitigation	What mitigation method was employed after or during the event (treatment, water treatment, monitoring, moving animals, etc.)?

**Table 2 animals-12-02423-t002:** Narrative summary of general study characteristics of 45 included cases.

Study Characteristic	Narrative Summary ^1^
Study type	The majority of papers were case reports, encompassing 71.1% of all included papers. Field observations, notes, surveillance studies, and epidemiological studies made up 15.4%, 4.4%, 4.4%, and 4.4% of included papers, respectively.
Location of incidents	North America, Africa, and Europe were the continents where cases were reported the most, contributing to 31.1%, 28.9%, and 22.2%, respectively. Cases in South Africa contributed to 30.8% of all African cases. The lowest number of cases occurred within Asia, contributing to 17.8% of cases. No cases of terrestrial wildlife were reported from Australasia or South America.
Presence of a harmful cyanobacterial bloom	Harmful cyanobacterial blooms were present in 64.4% of cases. When a bloom was present, the cause of the bloom was unknown in 51.7% of cases. Benthic cyanobacterial mats were also present in a number of cases and were the cause of intoxication in 8.9% of cases, all of which were lesser flamingo *Phoeniconaias minor* mortalities.
Cyanotoxin	Microcystin equivalents accounted for 64.4% of all cases included in this review. Of these, 56.9% were able to be identified to the variant level, which included MC-LR ^a^, MC-RR ^b^, MC-YR ^c^, MC-LF ^d^, and MC-LA ^e^, with MC-LR being the most common. Anatoxin-a accounted for 8.9% of all cases, and guanitoxin accounted for 2.2% of cases. Other toxins contributing to mortality, morbidity, or bioaccumulation accounted for 20.0% together, which included BMAA ^f^, Nodularin, DAB ^g^, and anabaenopeptides. Toxins were unknown in 13.3% of cases.
Cyanobacterial species	*Microcystis* spp. were the most prevalent freshwater cyanobacterial species at 40.0%. Alarmingly, 28.9% of cases were unable to identify the cyanobacterial species responsible. *Dolichospermum* spp. contributed to the second largest number of cases at 22.2%. *Oscillatoria* spp. were present in 13.3% of cases, and *Aphanizomenon* spp. were present in 6.7%. *Cylindrospermopsis* spp., *Arthrospira fusiformis*, *Nostoc* spp. and species in the order Stigonematales each contributed to 4.4% of cases. The remaining species encompassed cyanobacterial species, dinoflagellates, and raphidophytes, some of which were marine algal species. These remaining species were all only present in one case each and included *Plankothrix* spp., *Karenia brevis*, *Synechoccocus* sp., *Pleurochrysis pseudoroscoffensis*, *Chattonella marina*, *Gyrodinium uncatenum*, *Protoceratium reticulatum*, *Prorocentrum minimum*, *Phormidium terebriformis, Synechococcus bigranulatus, Spirulina subsalsa,* and an unknown species of the order Chroococcales.
Taxonomic groups of affected species	The class Aves made up the majority of cases at 48.9% of all cases, followed by Mammalia with 33.3%. Insecta, Reptilia, and Amphibia were present in only 11.1%, 8.9%, and 6.6% of cases, respectively. The order Anseriformes accounted for the most cases, present in 17.8% of poisonings. Carnivora and Artiodactyla from class Mammalia were involved in a number of cases, both present in 13.3% of cases. The order Phoenicopteriformes from class Aves also contributed to a large total with 13.3% of cases. Many other orders from classes Mammalia and Aves were involved in cyanobacterial poisonings, each included in one to five cases. Testudines constituted four out of the five cases involving reptilians. Additionally, both chronic and acute cyanotoxin poisonings were detected within Chiroptera, contributing to a total of 4 (8.9%) cases.
Environmental contaminants	Pesticides and heavy metals were both present in 8.9% of all cases and always occurred in the presence of each other. Sewage was present in 4.4% of all cases, and fertiliser was present in 2.2% of all cases. Only one case existed where environmental contaminants were known to have contributed to the case.
Mitigation measures	Mitigation measures were implemented in only 9 (20.0% of all cases) cases, and no investigations had mitigation in place already. Of the 9 studies implementing mitigation, 4 were successful, 2 were unsuccessful, and the outcome of mitigation was unknown in 3. Mitigation measures were not implemented or mentioned in 80.0% of all cases. Treatment and management of water was the most common mitigation measure, mentioned in 8.9% of all cases. The movement and translocation of animals and habitat management each occurred in 4.4% of all cases. Surveillance as an outcome was only reported in one case. Education and veterinary treatment were not reported in any of these wildlife reports.

^1^ Percentages may not total 100, as multiple characteristics are described in some publications. ^a^ Microcystin-LR, ^b^ Microcystin-RR, ^c^ Microcystin-YR, ^d^ Microcystin-LF, ^e^ Microcystin-LA, ^f^ BMAA, and ^g^ DAB.

**Table 3 animals-12-02423-t003:** Clinical diagnostics and investigative chemistry techniques used in 45 cases of cyanotoxin poisoning in terrestrial wildlife.

Method	Investigative Technique	Count (%)
Clinical diagnostic method	PME	19(42.2)
	Tissue sample (toxin identification)	17(37.8)
	Histopathology	10(22.2)
	Gut/stomach/faecal contents	10(22.2)
	Clinical signs	5(11.1)
	Blood sample	2(4.4)
Identification of cyanobacterial species	Microscopy	19(42.2)
	PCR	3(6.7)
Toxin presence and concentration	ELISA	18(40.0)
	HPLC (and multiple different detectors)	17(37.8)
	Bioassay (cow, mouse, catfish hepatocyte, or brine shrimp)	12(26.7)
	Mass spectrometry	11(24.4)
	Protein phosphatase inhibition assay	7(15.6)
	Cholinesterase inhibition assay	1(2.2)
	HNMR ^a^	1(2.2)
	TLC ^b^	1(2.2)

^a^ Proton Nuclear Magnetic Resonance; ^b^ Thin-layer Chromatography.

## Data Availability

Data presented in this study are openly available within figshare at https://doi.org/10.6084/m9.figshare.19919486 (accessed on 11 September 2022).

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
