# Peer review of "Reporting of Freshwater Cyanobacterial Poisoning in Terrestrial Wildlife: A Systematic Map"

_animals, 2022, doi:10.3390/ani12182423_

Round 1

Reviewer 1 Report

The review outlines the lack of evidence gathered with regard to wildlife intoxications by cyanobacterial blooms. The manuscript highlights that this is an often-overlooked area, with domestic animals and human health taking priority in previously published materials. The authors also demonstrate that there is a lack of a standardised methodology with which to determine the presence of harmful cyanobacterial species (in those studies focussed on wildlife intoxications), hampering efforts to compare events or build useful time series and trends. As such the authors seek to propose a recommended methodology for the use in further studies to aid with the collection of a holistic data set.

Overall, I found the paper to be well structured and clear. The paper was well written, with very few errors noted. I feel that the paper offers a valuable contribution, especially in highlighting the impact of cyanobacteria and the toxins they can produce on often overlooked populations of terrestrial wildlife. The systematic review approach is a good method for determining the current state of knowledge as well as allowing a more quantitative approach towards highlighting knowledge gaps and consistencies within the existing published literature.

There is a common theme of not distinguishing between cyanobacterial blooms and harmful algal blooms. As these groups of organisms are not related this is a distinction which should be made clear throughout the paper. I would recommend using a specific term for each with HAB properly referring to algal blooms and cyanoHAB or similar for reference to blooms caused by cyanobacterial species. Alternatively, I would like to see the authors define their own use of HAB at the opening of the manuscript, to ensure the reader is clear about the manner in which the term will be used throughout. Any mentions of microalgal blooms or species, especially those from marine environments, need to be properly prefaced with caveats as to their relevance to the main work of the paper. It may be useful to utilise examples from the microalgal toxin producers, but it must be made clear that similarities between these and toxic cyanobacterial species do not represent genetic relatedness so much as the potential for fulfilment of similar ecological roles.

I also feel that the recommended methods in this manuscript require more attention, the case for this is detailed more thoroughly in the specific comments.

Specific comments can be found below.

8 – Would prefer the term cyanobacterial blooms or cyanoHABs as the manuscript focuses on these rather than true microalgae. This will help to cement the focus of the paper. Additionally, recent synthesis of reports in the global HAB status report indicates no obvious global increase in HABS so if referring to microalgae then this should be considered in the introduction (Hallegraeff, G.M., Anderson, D.M., Belin, C., Bottein, M.Y.D., Bresnan, E., Chinain, M., Enevoldsen, H., Iwataki, M., Karlson, B., McKenzie, C.H. and Sunesen, I., 2021. Perceived global increase in algal blooms is attributable to intensified monitoring and emerging bloom impacts. Communications Earth & Environment2(1), pp.1-10.).

24 – Use of “harmful algal blooms” needs to be specific to cyanobacterial blooms, given the focus of the paper

47 – reference 3 is specific to marine HABs and so needs to be caveated – “This has been shown in the marine environment also, in species of harmful marine microalgae”

52&53 – toxicity only in bloom senescence?

79&80 – aquatic species more vulnerable to cyanotoxins than terrestrial? No citation

104&105- seems to contradict point in 79&80 suggesting widespread mortality in terrestrial species.

110 – cyanoHABs

114&115 – first mention of invasive species, a lack of baseline data, as already highlighted in the introduction, inhibits the assessment of invasive status for species of cyanobacteria

119 – Section incorrectly labelled as section 3 this follows on with the results section incorrectly labelled as section 4

187 – As far as I am aware PCR methods typically allow for the detection of the genetic sequences responsible for toxin production but not the toxins themselves. The detection of toxins would be more accurately undertaken via targeted chemical analysis such as LCMS/MS or immunoassay such as ELISA, as examples. Given that detection of the genes does not provide any information on the toxin profile or on the toxin concentrations this is an area I would like to see the authors expand to include in their recommended protocol for assessing toxicity events, especially as a range of proven analytical techniques exist for the determination of presence and concentration of cyanotoxins. Mentioned in Table 1 but does not form a component of the recommended protocol, other than being listed as investigative chemistry. This subject area has been widely investigated, a recent review (Sundaravadivelu, D., Sanan, T.T., Venkatapathy, R., Mash, H., Tettenhorst, D., DAnglada, L., Frey, S., Tatters, A.O. and Lazorchak, J., 2022. Determination of Cyanotoxins and Prymnesins in Water, Fish Tissue, and Other Matrices: A Review. Toxins14(3), p.213.) provides a valuable starting point for some of this information to be included in this manuscript.

211 – table 1 – correct “Was an algal bloom present” to “Was a cyanobacterial bloom present”

232 – replace “algal” from HAB

238 – replace reference to harmful algal blooms

238- table 2 – In section “Cyanobacteria species” multiple marine algal species are listed erroneously. Karenia brevis, Gyrodinium uncatenatum, Protoceratium reticulatum and Prorocentrum minimum are all Dinoflagellates, whilst Chattonella marina is a raphidophyte. These are all eukaryotic organisms and as such are far removed from cyanobacteria. Additionally, these would be considered marine species, yet the authors suggested that this environment was not considered in this review.

243 – figure appears to be relatively low resolution with some minor display issues apparent during review.

245 – no mention in figure legend of the meaning or characterisation of the green numbered circles appearing in the figure.

253 – section 4.3 (should be 3.3 see above) – no mention of the techniques used for the determination of toxicity, or toxin presence

313 – use of HABs, unsure if this reference is encompassing cyanobacteria and microalgae or if this is specific.

313 & 314 – two species are listed as invasive but invasive to where? The term invasive should be clarified so readers are aware of where the term has relevance. For example, in the citation 64, Cylindrospermopsis raciborskii (incorrect spelling in line 314) is listed as native to Indonesia-Java, and so would not be considered invasive in that region, potentially affecting its invasive status in the literature reviewed in the current manuscript.

384 – I would disagree with the recommendation of exclusively utilising PCR for the detection of harmful cyanobacterial species and toxins. There is strong support for the use of PCR and qPCR in the literature for the detection of cyanobacterial species and toxin producing genes but the direct detection of cyanotoxins is a field dominated by immune assays and analytical chemistry. I have not found indication of a proven method of immuno-PCR which would facilitate the detection and quantitation of cyanotoxins, within the literature. Indeed, the reference used in line 387 (reference 85), offers considerable evidence to suggest that there is not always a sufficient positive correlation between PCR analysis of toxigenic species or toxin synthesis genes, and actually determined toxin levels. I do not feel that the authors present a strong enough case for the use of PCR for toxin detection, as they do not make extensive mention of the relevant techniques, nor do they offer considerable sources demonstrating this atypical use of PCR.

438-440 – reference 91 makes no mention of PCR or ELISA, the authors should check citations carefully to ensure all literature is correctly cited in the bibliography. This citation should be amended or removed.

488 – edit reference to harmful algal blooms to better reflect the focus on cyanobacteria

Reviewer 2 Report

The authors present a review on understanding how to report cyanobacterias poisonings in terrestrial wildlife. Although an interesting topic, unfortunately the review has a number of critical failures that do not warrant publication.

1. The authors recommend using PCR methods to understand toxicity events. This is an inappropriate method as PCR only measures the genes for toxin production, not the toxin analysis itself. Therefore, only the potential for toxin production can be ascertained. Even if cyanobacteria are present in stomach contents, without an assessment of toxins in target organs, no conclusions can be drawn.

2. Search terms: Why were no terms such as "intoxication", "death", "poisoning" etc included in searches?

3. Table 2: Anatoxin-a/a(s) should be separated into two toxins. They are very distinctly different and anatoxin-a(s) is now named guanitoxin.

4. Table 2: cyanobacterial species. Dolichospermum is now the new name of the genus Anabaena. Furthermore, Pleurochrysis, Chattonella, Gyrodinium, Protoceratium and Prorocentrum are not cyanobacteria.

5. Lead author affiliation section: why was this included? This seems redundant.

6. Line 294-5 and line 379: This should be changed around. You agreed with the authors use of methods, not the other way around.

7. Line 309: Microcystis and Anabaena are freshwater cyanobacterial genera and are unlikely to be transferred in ballast water.

8. Please check references: Line 321, ref 65 is wrongly cited and line 310, ref 62 is wrongly cited. Please check the others.

9. Line 375: PCR and qPCR cannot detect toxins only the genes for their production.

10: Line 438-440. ELISA can detect toxins, PCR cannot.

Reviewer 3 Report

Reporting of Freshwater Cyanobacterial Poisoning in Terrestrial Wildlife:  A Systematic Map

Ash and Patterson

This is a very interesting and needed paper detailing wildlife poisonings by cyanobacterial toxins. However, the authors emphasize the faults of the reports and really don’t offer ways to address the problems beyond saying “do more tests.”  I think the summaries are good, but the categories are not well-defined and I wasn’t always sure what they meant.  It would help to let readers know the denominators of your percentages.

I suggest the authors acknowledge that, when the older events were investigated and reported, many of the techniques recommended in this paper simply didn’t exist. Taxonomy is fluid, and there are many, many more toxins than we have methods to look for. I suggest adding a section on “problems with early reports” and then offer the list of recommendations based on the current science and then re-orient the paper to “this is how we can do this better.” I think the paper would be more succinct and an even better contribution to the literature.

The paper often mentions human, wildlife, and ecosystem health.  However, it is not only wild animals that are affected.  Sick pet dogs are often one of the first indications that we have a toxin-producing bloom. I suggest that you modify your list a bit:  human, animal (including wildlife), and ecosystem health. I saw that you did that later, but putting it up front is useful.

Specific comments:

Line 47. We’ve seen cyano blooms that color the water bright blue or yellowish. Suggest modifying this to …scums that can be green, bright blue, or other colors.”

Line 49. I can’t tell what “This results in..” is referring to.

Line 64. Please provide a reference for the statement about BMAA as the reported effects of exposure are not universally accepted.

Line 67.  I disagree with the statement “Whilst the threat….well known and largely understood….”

There is much we don’t know about the specific human health risks posed by these toxins. At this point, we do not have solid case definitions for human health effects.  I suggest you modify that sentence to reflect the uncertainties. And I agree that the impacts on wild animals is severely understudied.

Line 84. The relationship between ALS and cyanobacterial toxin exposure via the food chain is highly disputed (see review by Chernoff et al.—link below), particularly because “ALS” is used to describe a wide array of neurologic diseases that may or may not be related. I suggest you delete that as it doesn’t impact your message.

To link to this article: http://dx.doi.org/10.1080/10937404.2017.1297592

CDC has an active One Health Harmful Algal Bloom System that allows states to report cyanobacterial toxin-related illnesses in animals of all kinds, people as well as information about the blooms themselves.  Perhaps this would be useful as you create a surveillance system. Harmful Algal Bloom–Associated Illnesses | CDC

Line 139. Just curious, why did you not include “bloom poisonings” in your search

Line 159. You don’t need the “were included” at the end of the sentence.

Line 165. There is an incomplete sentence.

In Figure 1. Can you explain the reasons used to exclude reports?  For example,  you note above that study type is not very limited, but 23 reports were exclude for “study type.”

In Table 2 in the row with cyanotoxins.  Please define your acronyms.

In Table 2 in the row for species. What does “These” stand for in the last sentence. I think those are all marine species, but the text isn’t clear about what you mean.

In Table 2 in the row for environmental contaminants:  You note that pesticides and heavy metals were present in 8.8% of cases.  Is that all cases or just those where they were tested for? I have the same question for sewage and fertilizer.  It would help your readers if you clarify this.

In Table 2, row with mitigation measures. For the subcategories beginning with treatment and management, are the percentages from all cases or just the 9 where mitigation was involved?  Please clarify.

Perhaps you could say:  “Education….were not reported…. Rather than “not employed.”

Figure 4.  The link did not produce an interactive map, but that may just be my system blocking it.

Line 291. Suggest rewording of the last sentence.  “….only one case, in South Carolina, United States, reported an invasive…

Line 297. I believe it is important that some of your cases involved both wild and domestic animals.  Can you provide the species of domestic animals affected?

Table 3, row with clinical diagnostic method: have you defined PME somewhere?

Table 3, row with Toxin presence and concentration.  Can you explain a cow bioassay? The mouse bioassay results in the death of the mouse, what happens to the cow?

Please define HNMR, TLC

Line 326.  I am not sure “encompass” is the right word.  Maybe “represent?”

Line 330.  Earlier, I assumed that the “active investigations” study type meant some ongoing activity triggered by a bloom or illness. I suggest you provide more information about how you defined “active” and “retrospective” investigations.

Line 335.  You do not note issues with species identification until this mention of an invasive species.  Do you have information to suggest that species are misidentified?

Line 340.  Do you mean that they tried to identify the cyanobacteria but were unable to or that they didn’t try?  That’s important to what you are saying about invasive species.

Section 4.1  There are a lot of ideas here and it’s hard to follow. Can you rearrange the text to make a story that’s easier to follow?  Also, the fact that your reports don’t mention some species could be related to the date of the reports.  New species and new toxins have been identified, and new methods developed over the last two decades, and some investigators may not have had the tools needed to make these assessments. You could make a recommendation that, in creating these reports, the investigators do a thorough search of available analytic methods and the most current taxonomic names (as these change over time).

Line 363.  Is it true that investigators will analyze samples for toxins without first identifying the organism and then doing the appropriate chemical analysis based on the toxins these organisms produce? Investigations like this are often resource-restrained, and I wonder if there would be a scattershot approach to toxin identification without knowing first what the organism is. Can you clarify, please? Also, depending on the timing of the event, we found that agencies may “look under the lamppost,” that is, they will test using inexpensive methods such as ELIZA and ignore the possibility that other toxins that cannot be analyzed with ELIZA kit available will be present.

Line 378. I would argue that chronic exposure does not always involve bioaccumulation. It could also arise from repeated exposures that inflict repeated injury without bioaccumulating. I don’t think this statement, although identifies one issue with chronic exposures, really summarizes the issues in documenting these exposures.

Line 393. You don’t explain how you arrived at the conclusion that environmental contaminants and cyanotoxins contributed to mortality in the one case.  That would be helpful here.

Also, because the other reports did not note the presence of environmental contaminants does not mean they weren’t present. It could be that preexisting conditions related to chronic exposure to other contaminants tipped the scale for the animals to make them more susceptible to the effects of cyanotoxins, maybe even at levels that would normally not be fatal.

Line 399.  Do you mean detection and surveillance of disease?  It would be helpful to note that.

Line 400. What do you mean “in the absence of wildlife health surveillance”? How were the illnesses/mortalities found? Could the surveillance be casual rather than captured in some official system?

Line 403. Note that the WHO guidance values for microcystins are based on lifetime exposure.  You would want guidance for more acute exposures in animals as these events may happen seasonally (e.g., during the dry season).

Line 407. By “parties” do you mean both humans and animals? I suggest spelling that out.

Line 414.  I don’t think anyone has proposed that microscopy alone can distinguish toxic from non-toxic cyanobacterial strains. I think calling it “unreliable and inconsistent is not fair to the purpose of the methodology (i.e., to identify species and sub-species).

Line 417.  The EPA does have some recommended toxin detection methods. And, I don’t think your analysis identified that as the reason for lack of analysis. I wonder if it’s really more a lack of resources?  Unless it is mentioned in the reports, I suggest that you not note the lack of a “gold standard” analytic method as the reason there was no toxicity testing.

Line 462. Do you meant that the species should be monitored to detect potentially harmful blooms before they become a wildlife health risk?

Line 470.  I suggest including public health professionals/veterinary public health professionals

Line 474.  See the U.S. EPA’s Cyanobacteria Assessment Network (CyAN). Perhaps you could use this as an example of how to monitor for blooms? I don’t know if the satellite data cover more than the U.S., but there may be other possibilities and the technology, data, and interpretation are already in place; oceancolor.gsfc.nasa.gov/projects/cyan/

Line 522.  You can delete the last sentence as this is not a limitation.

Line 528. We really do have  a lot of information on about disease surveillance and mitigation of blooms.  I think the problem is more lack of interest/resources and conflicting priorities.  Can you add some text suggesting why addressing cyanoHABs should be a priority, particularly for developing nations?  That would strengthen your argument.

Round 2

Reviewer 2 Report

The authors present a corrected review concerning wildlife poisonings with cyanotoxins. Although an interesting topic, this reviewer is not sure what this review adds to the field.

One issue concerns the authors claim that prior studies agreed with their recommendations throughout the manuscript. This is not true as the authors agreed with those prior studies, not the other way around. Furthermore, the recommendations are based on a previous manuscript published by Kaushik and Balasubramanian (no reference number on line 192). Although much improved, this reviewer does not feel that this manuscript should be published. Furthermore, there are a number of specific points that require attention.

1. Line 34-35: Investigations have to occur after ingestion/mortality has occurred. It can't be done the other way around.

2. Line 47: cyanoHABs should be capitalized.

3. Line 48: Cyanobacteria are generally described as a blue-green scum.

4. Line 53: should say "species such as those in the dinoflagellates and diatoms" - diatoms and dinoflagellates are not species.

5. Line 90: No reference numbers or references for Skar et al. and Cox et al. Also, which reference discussed microcystins in bats? In addition, the work of Cox et al. was in flying foxes. Are these two papers studying the same organisms?

6. Lines 414-5: Many studies have also used microscopy followed by toxin analysis which is completely acceptable in such scenarios.
